# Technological Platform for Vertical Multi-Wafer Integration of Microscanners and Micro-Optical Components

**DOI:** 10.3390/mi10030185

**Published:** 2019-03-13

**Authors:** Sylwester Bargiel, Maciej Baranski, Maik Wiemer, Jörg Frömel, Wei-Shan Wang, Christophe Gorecki

**Affiliations:** 1Département MN2S, FEMTO-ST (UMR CNRS 6714), University of Bourgogne Franche-Comté (UBFC), 15B Avenue des Montboucons, 25030 Besançon, CEDEX, France; sylwester.bargiel@femto-st.fr (S.B.); maciej@smart.mit.edu (M.B.); 2System Packaging Department, Fraunhofer Institute for Electronic Nanosystems (ENAS), Technologie-Campus 3, 09126 Chemnitz, Germany; Maik.Wiemer@enas.fraunhofer.de (M.W.); joerg.froemel.e5@tohoku.ac.jp (J.F.); Wei-Shan.Wang@infineon.com (W.-S.W.)

**Keywords:** MOEMS, micro-optics, vertical integration, multi-wafer bonding, confocal microscopy

## Abstract

We describe an original integration technological platform for the miniaturization of micromachined on-chip optical microscopes, such as the laser scanning confocal microscope. The platform employs the multi-wafer vertical integration approach, combined with integrated glass-based micro-optics as well as micro-electro-mechanical systems (MEMS) components, where the assembly uses the heterogeneous bonding and interconnecting technologies. Various heterogeneous components are disposed in vertically stacked building blocks (glass microlens, MEMS actuator, beamsplitter, etc.) in a minimum space. The platform offers the integrity and potential of MEMS microactuators integrated with micro-optics, providing miniaturized and low cost solutions to create micromachined on-chip optical microscopes.

## 1. Introduction

Born from the field of micro-electro-mechanical systems (MEMS) and micro-optics, first the Optical MEMS and then the micro-opto-electro-mechanical systems (MOEMS) technologies offers attractive and low-cost solutions to a range of device problems that require both the complex optical functionality and high optical performance at a low cost [1,2]. Thus, MOEMS solutions include optical devices for telecommunication, sensing, and mobile systems, such as gratings, shutters, scanners, filters, micromirrors, switches, alignment aids, microlens arrays, optical filters, as well as hermetic wafer-scale optical packaging. The heterogeneous integration technologies at the wafer-level for MOEMS [3,4] permit the combination of the dissimilar classes of materials and components into single optical microsystems. Thus, high-performance subsystems can be associated by using each technology where it is the best and in a way that would otherwise not be possible to integrate, thereby forming complex and highly integrated microsystems. On the other hand, the lack of miniaturization in conventional three-dimensional (3D) imagers limits their application fields. Micro-optical and micro-mechanical systems represent the huge potential of miniaturization, putting together the potentialities of optics and MEMS and offering an attractive integration platform. Here, one of the most critical obstacles to achieve cost-effective solutions is to successfully integrate the optical microsystems that require numerous hybrid technologies, to assemble them within tolerances of few microns, and to deal with packaging issues.

The purpose of this contribution is to propose solutions to the technological issues of vertically integrated optical micro-instruments, which are fabricated by combining heterogeneous technologies (glass, silicon technologies). Our goal aims to propose a new generation of on-chip microscopes, such as those described in Ref. [5]. The miniaturization of optical imagers with MOEMS technology is a challenging task [6,7,8]. It implies not only scaling down of the individual system components, but also a simplification of their arrangement, and, what is of great importance, the adaptation of optical system design to the specific constraints of the chosen technology. In the ideal case, one would imagine batch fabrication of a complete functional instrument on-chip in the continuous wafer-level process flow, taking advantage of 3D vertical integration methods. According to this approach, several wafers of different functionality (e.g., microactuator wafer, micro-optical wafer, separator wafer, photodetector wafer, etc.) are vertically aligned and then bonded to achieve mechanical and electrical connections, followed by the dicing of wafer stack to obtain individual MOEMS devices.

An example of the 3D device that is based on such an approach of fabrication is the vertically integrated 3D optical microscanner shown in Figure 1, which is suitable for the implementation of the miniature confocal microscope [9]. The microscanner includes a micro beam-splitter and a microlens doublet to respectively implement the collimation and scanning of the light beam issued from an external light source. We describe several technological solutions, further referred as to “technological platform”, which is developed to support the wafer-level batch fabrication of this microscanner for on-chip microscopy, based on free-space glass micro-optics and MEMS. The architecture of this microscanner, which implements the vertically stacked building blocks, is described in Section 2 of the present contribution. In Section 3.1, series of original key microfabrication processes of micro-optical components are described. Section 3.2 demonstrates that these microoptical components can be successfully integrated with MEMS actuators to build the 3D microscanner of Figure 2. In Section 3.3, a study of the accurate positioning, bonding, electrical interconnecting, and reliable packaging of building blocks is detailed. 

## 2. Method: The Optical Design of 3D Micro-optical Scanner

Figure 2a shows the details of construction of the 3D micro-optical scanner presented without the glass micro beam-splitter. Here, the vertical integration scheme permits the integration of cascaded microlenses on the top of two scanners. The actuated optics are represented in Figure 1 by plano-convex microlenses, which is an ideal situation. Because of the difficult integration of such microlenses on the top of thin actuators, the preliminary version of the 3D microscanner of Figure 2a includes two glass ball microlenses of 500 µm in diameter. All of the building blocks of the microscanner are mechanically and electrically connected at the wafer-level by anodic bonding. Two electrostatic microactuators are used for microlens displacement. The *x-y* microactuator is a translation micro-stage with frame-in-the-frame architecture [10,11], employing four comb-drive actuators and eight straight spring suspensions. The device provides an independent movement of microlens along the *x* and *y* directions in the range of ±35 µm with resonance frequencies that are in the range of 600–700 Hz (depending of the spring design) and for a driving voltage below 60 V. The z microactuator is an electrostatic parallel-plate microactuator that ensures the vertical displacement of microlens in the range of ±20 µm at the resonance frequency of around 600 Hz and a driving voltage below 70 V. Two glass lids in borosilicate glass ensure excellent optical transparency through the whole stack for incoming illumination (laser beam), providing, at the same time, hermetic sealing of the microsystem. Thus, the sensitive microactuators and glass microlenses are protected from environmentally-induced degradations (dust, moisture). 

The distance between scanning microlenses has to be controlled within, ideally, tolerances below 5 µm. As described later in the Section 3.3, this tolerance value is challenging, while taking into account several limitations that result from the wafer-level fabrication principle (e.g., thickness dispersion of separator wafer). In this work, carefully selected mechanical spacer wafers (thickness tolerance ±5 µm), made of low temperature co-fired ceramic (LTCC) from NIKKO, were used. Due to the matched coefficient of thermal expansion (CTEs), this material can be anodically bonded to silicon and, moreover, it permits the integration of Au-based vertical electrical connections. The latter feature is employed to drive z microactuator, sandwiched between glass and ceramic components, by fabrication of the electrical connections through a stack of one glass wafer, two silicon-on-insulator (SOI) wafers, and one ceramic wafer, as described in Ref. [12]. 

Figure 2b shows the optical design of 3D microscanner, presenting strong correspondence with the stacking architecture of Figure 2a [13]. The equivalent optical system is composed of three blocks: a collimation block, a scanning block, and a focusing block. This includes two glass ball microlenses, which are hybridly assembled with actuators as an afocal scanning doublet. This doublet performs the *x-y* and z-scans, where one monolithically integrated plano-convex glass microlens is responsible for the generation of a focused spot at the output of the optical system. The working distance is 160 µm, thanks to the numerical aperture of the glass ball (NA = 0.45), providing a diffraction-limited focal spot that defines the scanning volume of the 3D microscanner. 

The aberrations due to the use of glass ball microlenses limit the optical performances of microscanner. In particular, for large lateral lens displacements, the off-axial aberrations (coma, astigmatism) deform the focal spot, resulting in a different resolution along the two perpendicular directions, as shown in Figure 3a. The lateral scanning is linear, with the scan displacement Δ*y* and being coupled to the axial scanning, which is slightly nonlinear. This combined motion of scanning optics lead to the creation of a scanned volume of Figure 3b, which is located in the vicinity of the focus point of the focusing microlens. 

The expected optical parameters of microscanner, defined within the scanning volume shown in Figure 3b, are listed in Table 1. 

The 3D microscanner is designed to operate at resonance modes of both microactuators to perform the scanning based on 3D Lissajous patterns. In such a scanning mode, the scan resolution is defined by the frequencies ratio and relative phase shift [14,15]. Here, the best solution consists of synchronizing all axial phases (ϕ_x_ = ϕ_y_ = ϕ_z_) and controlling the scanning pattern only by adjusting the driving frequencies. Indeed, by choosing appropriate frequency ratio of *f*_x_/*f*_y_ and *f*_x_/*f*_z_ the density of the scanning line can be adjusted to match the desired scan resolution. In practice, the driving frequencies should be close to the resonance frequency of the individual microscanners, however the finite quality factor of the scanners structures allows enough space for adjusting the driving frequency, while still allowing operation in the resonance mode. The examples of 3D Lissajous patterns, as presented in Figure 4, were obtained with scanning frequencies that were fixed at *f*_x_ = 650 Hz and *f*_z_ = 452 Hz, while all phase shifts were set to zero. 

Due to the curved shape of scanning volume that is shown in Figure 4b, a certain coupling between lateral and axial scanning is observed. The 3D scanning Lissajous curve, representing the path of the focalized laser spot, is generated by harmonic oscillations of the three-axis scanner. The geometrical transformation was done within paraxial approximation, where impact of curvature was approximated by a parabolic deformation, leading to:(1)$$z_{spot}=z-\frac{x^{2}+y^{2}}{2R_{img}}$$
where x, y, and z represent scanners positions, R*_img_* is the optical curvature of the image plane that was estimated to 35 μm, and *z_spot_* is the coordinate of the focus position. 

The lateral scanning coupling was also analyzed within a paraxial approximation that leads to a linear transformation: (2)$$x_{spot}=x\left(1+\frac{z}{\alpha }\right)\;\mathrm{and}\;y_{spot}=y\left(1+\frac{z}{\alpha }\right)$$
where α is the coupling coefficient, found to be 80 μm.

The scan control relaying on frequencies control is much easier to implement than the classical linear scanning system. The first important advantage of all-resonance scanning is that it does not require a complicated feedback control system. Secondly, it employs natural dynamical mode of the microscanners that permit a very stable and repeatable performance. In addition, the third is that it allows for reaching maximal displacement amplitudes that are accessible to the scanners in the very efficient way of limiting the values of driving voltage. 

The scanning performance [13] mainly depends on the ratio *f*_3_/*f*_2_, where *f*_2_ represents the focal length of the scanning lens, while *f*_3_ is the focal length of the focusing lens. Thus, to amplify the scan amplitude *f*_3_ should be selected higher than *f*_2_. The scanner’s NA defines the optical resolution, which is proportional to *f*_2_*/f_3_*, corresponding to the inverse of the dependence of the scanning performance:(3)$$NA\propto \frac{1}{f_{1}}(\frac{f_{2}}{f_{3}})$$
where *f*_1_ represent the focal length of the collimating lens.

In practice, Equation (3) means that it is impossible to maximize both the scanning volume and the resolution at the same time using only the ratio *f*_2_*/f*_3_. 

All of the optical elements in the microsystem contribute to the spherical aberration, degrading the optical resolution. Other factors that reduce the optical performances appear when the lateral scan of scanning lenses is operating. In Figure 5, we evaluated the contrast transfer function for three positions of the scanner: the axial position where all the lenses are aligned on the optical axis and two lateral scanning positions: Δ*y* = 17.5 μm and Δ*y* = 35 μm. 

Resolution in the on-axis case is reasonably close to the aberration-free system (i.e., diffraction limited). However, lateral scanning visibly deteriorates the system performance. In particular, for large lateral lens displacements, the off-axial aberrations (coma, astigmatism) deform the focal spot, resulting in a different resolving power along the two perpendicular directions. This resolution is always worse along the scanning direction.

## 3. Results and Discussion

### 3.1. Glass Microoptical Components: Overview of Key Processes

#### 3.1.1. Characterization of Scanning Glass Ball Microlensess

The preliminary version of microscanner employed glass ball microlenses made of N-BK-7 glass (Edmunds Optics) that were integrated on the scanner by hybrid assembly. We evaluated the optical properties of these microlenses by the direct scanning of the volume of Intensity Point Spread Function (IPSF) [16]. The measurements consisted of sequentially recording the images of lateral slices of the scanned 3D focus volume that is generated by the investigated micro-optical component. Figure 6a shows a slice of measured IPSF of a ball microlens (NA = 0.45 and 500 µm in diameter), as measured at the focal plane. Figure 6b shows the axial cross section of IPSF, measured by the displacement along the optical axis, where the spherical aberration that is generated by the ball lens is well visible in the left side of the image. This effect is in agreement with theoretical calculations. 

The glass ball microlens is assembled onto the scanning microactuators by means of pick-and-place principle and the localized bonding technique, as described in Section 3.2.3. Here, the main advantage to use such components is that the hybrid ball-microlens assembly prevents the integrity of MEMS actuators and permits the easier prototyping of a low-resolution 3D microscanner. 

#### 3.1.2. Fabrication of Focusing Plano-Convex Glass Microlenses by Micro-Blowing Process

To produce the conical and spherical shapes in borosilicate glass, which is necessary in the fabrication of the microoptical components bondable to the silicon, we used the technique of glass-blowing. Figure 7 shows that the micro-blowing process that is based on the anodic bonding of a borosilicate glass wafer (Borofloat 33, Schott AG, Mainz, Germany) on the top of a silicon cavity, performed at well-controlled pressure into the cavity. It is followed by the thermal reflow in a atmospheric furnace leading to local deformation of the glass. If the bonding is performed under vacuum, the deformation of the glass fills the cavity with a spherical shape. This technique, completed by the polishing of the top side of glass wafer and releasing of silicon, is used to fabricate the plano-convex microlenses that are described in Ref. [17]. If the bonding is performed at the atmospheric pressure, it produces a piston effect that deforms the surface of the glass towards the top of the cavity and generates a conical cavity, which is used to fabricate the micro-axicons [18] that are necessary for the generation of Bessel beams.

The fabrication method of plano-convex microlenses is based on a non-contact thermal reflow of glass material in vacuum, over a micromachined cylindrical cavity, as shown in Figure 8a [19]. First, the cylindrical-shaped cavities with desired diameter and depth are formed in a silicon wafer (step 1) by the deep reactive-ion *etching* (DRIE), playing the role of the lens molds. Subsequently, a standard 0.5 mm-thick Borofloat 33 wafer is anodically bonded to the silicon mold wafer in vacuum (step 2). Afterwards, the silicon/glass structure is heated up to the temperature between the annealing and the softening point of the Borofloat 33 glass, i.e., 560 °C and 820 °C, respectively (step 3). Due to the pressure difference between the cavity and the atmosphere, the glass flows into the cavity, creating a lens with an approximately spherical profile and an extremely smooth surface (Ra < 1 nm). In order to obtain the required sag of the lens, the time of thermal reflow process is controlled, whereas the other process parameters, such as the bonding vacuum and reflow temperature, are constant. Since the top surface of glass becomes concave after reflow, glass planarization procedure (lapping and polishing) is performed, as shown in Figure 8b. Figure 8c reflects the resulting array of microlenses after selective wet etching of silicon wafer in hot potassium hydroxide (KOH) solution. The main advantage of this fabrication process is its natural ability to form glass lenses with an excellent surface quality (because it is a surface tension controlled process) in a wide range of diameters from tens of micrometers up to 2 mm.

The comparison of aberration effects between the ball microlenses and plano-convex microlenses demonstrates a better performance for plano-convex microlenses. In particular, the comas and astigmatism terms are divided by a factor of 10 when compared to a ball microlens. A better lateral resolution is obtained with plano-convex microlenses with a range of 2.3–2.9 µm within a bigger scanning volume (80 × 80 × 60 µm^3^). This technology is a good alternative to replace the ball microlenses in the future. However, the corresponding axial resolution is limited to 20 µm. An additional advantage is that this technology is well adapted for array-type microscopes. The optical performances are better for the plano-convex configuration, but these glass microlenses are more difficult to integrate with MEMS actuators—the thermal reflow of is a source of residual stress. 

#### 3.1.3. Fabrication of Glass Micro Beam-Splitter by Saw Dicing

The conversion of a 3D microlens scanner into a fully integrated confocal microscope requires the fabrication of a micromachined planar micro beam-splitter [20,21] to redirect the light beam that returns from the sample to the detector. We have developed a micro beam-splitter based on the saw dicing of Borofloat glass, as described in Ref. [22]. 

Figure 9a presents the construction of a planar 50/50 micro beam-splitter based on 45° saw dicing of Borofloat 33 glass wafers and PECVD deposition of dielectric thin film (SiOxNy) and metallic micromirror layers. When compared to typically used miniature beam splitting cubes (<5 mm^3^), this micro beam-splitter offers wafer-level integration and a compact design (<1 mm^3^), which are compatible with anodic bonding assembly. Moreover, due to embedded micromirrors, it allows for out-of-plane reflection of back-reflected sensing beams. The formation of 45°-inclined facets (see Figure 9b) in glass was performed by using an automatic high precision dicing saw DISCO (model DAD321), operating with custom-made dicing blades with polishing properties. The dicing procedure was investigated in terms of optimal dicing parameters and the cleaning procedure of the blade, resulting in a low roughness of diced surfaces (Ra < 10 nm). The optimal dicing speed was found to be 0.2 mm/s at a medium spindle revolution speed of 20,000 rpm. Figure 9c shows the light path when the incident light hits the semi-transparent stack and it is reflected by the mirror facing the photodetector. 

From the normally incident beam, the multi-layer stack (Fresnel reflections subtracted) transmits 48.5% of TE-polarized light, i.e., close to the designed 50% splitting ratio. This value demonstrates the correct behavior of the multi-dielectric stack. The measured reflected beam is equal to 25.3% of the incident light. These measured efficiencies lead to a beam splitting ratio T/R = 65.7/34.3 at λ = 632 nm with 26.2% losses. Hence, future works need to focus on decreasing the optical losses by further improving the surface quality of the diced glass facets. The complete integration of the micro beam-splitter with the MOEMS scanner is schematically shown in Figure 1. One of the possible integration solutions would be to create this block as a separate sub-component that can be fabricated with a standard process at the wafer level, while assembly and bonding with the microscanner could be performed at the chip level by means of e.g., flip-chip bonding.

### 3.2. 3D MEMS Scanner with Integration of Glass Ball Microlens

Section 3.2 focuses on the realization of the scanning MEMS actuators and the assembly of glass ball microlenses on the top of actuator membranes.

#### 3.2.1. Electrostatic Microlens x-y Scanner

Numerous microscanners that are based on the comb-drive technology have been developed [23,24]. In our case, the design, fabrication, and characterization of the *x-y* scanner are detailed in Ref. 10. However, the version of scanner that is presented here is modified in order to be integrated with a *z*-axis scanner and adapted to additionally act as a supporting holder of the ball microlens. Figure 10a shows the *x-y* microscanner that consists of a suspended central platform, which is connected by a system of spring suspensions to two frames (outer support frame and inner movable frame), and driven independently following two axes by four lateral comb-drive actuators. The outer support frame contains the first pair of comb-drives (Comb1, Comb2) that provide the displacement of the inner frame along the x-axis, as shown in Figure 10b. The second pair of comb-drives (Comb3, Comb4), attached to the inner frame, only displaces the platform along the *y*-axis. The central platform has a circular aperture, which is necessary for the integration of a glass microlens. The straight spring suspensions were designed to decouple the two lateral directions of actuation (*x-y*) without mechanical interferences, i.e., to be very compliant along the required movement direction, while being very stiff in other directions. This ensures the transverse stability (pull-in) up to 150 V of driving voltage. The ratio of required displacement (±35 µm) to spring length (1.55 mm) ensures the linear behavior of springs. The expected micromechanical parameters of the *x-y* scanner are summarized in Table 2.

The fabrication flow-chart of 16 main steps is summarized in Figure 11. The *x-y* scanner was fabricated on a 4”, p-type, (100)-oriented SOI wafer, including a 30-μm thick device layer (DL), a 400-µm thick substrate layer, and a 1.5-µm thick buried oxide (BOX). The SOI wafer is processed from both sides to form the structures of all silicon actuators in the DL, as well as to form the through-wafer electrical interconnection structures. The vertical SiO_2_-filled trenches in the DL allow for electrical insulation of different parts of the actuator, as well as mechanical connection to the suspended structure. The process includes five main phases: (i) fabrication of vertical SiO_2_ insulation trenches (VIT) in the DL, (ii) fabrication of through-silicon-vias (TSV) interconnects in the handle layer (HL), (iii) transfer of the comb-drive pattern into the thermal SiO_2_ layer and KOH etching of TSV ports in the DL; (iv) fabrication of comb-drive actuators; and, (v) releasing process in vapor HF. 

All the metallic pads for electrical contacts and the sealing frame for Au-Au thermo-compression bonding are located on the top side of the SOI wafer, on the external frame of the actuator. The etching of high aspect ratio structures was performed while using the DRIE Bosch process. We used the vapor HF to etch the BOX layer in the releasing step of the fabrication process. In the last fabrication step, the glass ball microlens is assembled and bonded to the silicon platform. 

Figure 12 illustrates the fabrication steps of the vertical insulation trenches. The method is based on the complete thermal oxidation of narrow silicon trenches (1.7-μm wide, aspect ratio of 18), etched through the whole thickness of the DL by an optimized DRIE etch (SF_6_/C_4_F_8_ chemistry), and Cr hard mask. In order to compensate the underetching of the Cr mask, much smaller 0.8-µm wide openings have to be initially created by the stepper machine. After the wet thermal oxidation process, the total thickness of vertical SiO_2_ wall is ~3.4 μm, resulting in a ~2-μm thick oxide on the surface. 

Figure 13 illustrates one critical steps of fabrication, which consists of the transfer of a precise microactuator pattern into the SiO_2_ hard mask layer. This requires high quality photolithography as well as a suitable dry SiO_2_ etching method, providing minimal underetching of the mask layer. The transfer of the comb-drive pattern into the 1.6-µm thick SiO_2_ layer through plasma etching was made while using a ICP DRIE machine (SPTS Rapier) with C_4_F_8_/Ar/O_2_ chemistry through the spin-coated photoresist mask. The measured gap between fingers in the SiO_2_ layer was 3.9 µm, indicating an underetch of 0.6 µm on each side. In the last steps, the scanner structure is released by an etching of the BOX layer in vapor HF (Idonus), followed by the assembly of the ball microlens on the movable platform of the *x-y* scanner and laser bonding (step 16), according to the procedure described in Section 3.2.3.

#### 3.2.2. Electrostatic Microlens z-Scanner

The construction of the *z*-axis scanner is based on the electrostatic parallel-plate actuation principle and it employs the customized SOI wafer with a 15-µm thick DL and embedded actuation cavity, as shown in Figure 14a. The movable platform of the scanner, formed after DRIE of the DL, represents a movable electrode of the electrostatic microactuator, whereas the surface of anisotropically etched cavity then plays the role of a counterelectrode. The 4 × 4 mm^2^ platform is suspended 80 µm (initial gap) over the counterelectrode by four multi-folded spring suspensions (fixed-guided beams). Due to their robust and compact construction, symmetry, low stiffness in the movement direction, as well as ability to relieve compressive loads, this type of springs is employed. The springs are located either externally to the platform or inside of the platform, as shown in Figure 14b. The latter allows for decreasing the footprint of the device. The expected micromechanical parameters of the z-scanner are summarized in Table 3.

The fabrication of z-scanner requires a “home-made” customized SOI substrate with a large (4 × 4 mm^2^) and 15-µm thin membrane that is suspended over an embedded actuation cavity with through-hole for light transmission. For that purpose, 80-µm deep cavities are first etched into a 400-µm thick Si wafer in KOH solution (10 M, 80 °C) while using a thermal SiO_2_ hard mask. Next, a direct bonding process against a “sacrificial” SOI wafer (device thickness of 15 µm, BOX thickness of 0.5 µm, and 400-µm thick handle wafer) takes place. For this direct bonding, a wet pretreatment, followed by a vacuum bonding process and an annealing step, are applied. The pretreatment consists of successive cleaning steps using the standard solutions (RCA1, RCA2, and again RCA1). The bonding is done at low pressure (<10^−4^ mbar) using a standard bonding equipment. For the annealing step, the parameters are 800 °C, 6 h in nitrogen in a horizontal furnace. The quality of the direct bonding has been monitored by an infrared microscope, showing only a few voids at the wafer edge, as shown in Figure 15a. The subsequent processing of the sacrificial SOI, based on KOH thinning of its HL, leads to the transfer of its DL to the final customized SOI with embedded cavity. The through-holes are then fabricated from the backside by DRIE etch. An optical microscope image of the resulting final wafer is shown in Figure 15b. It can be noted that the large membranes protrude slightly above the surface due to intrinsic stress. To minimize this deformation, the wafers were additionally annealed at 250 °C, before being processed.

The subsequent process that is necessary to finalize the fabrication of the entire microsystem includes five main phases: (i) fabrication of TSV ports in the DL by the KOH etching; (ii) fabrication of TSV contacts by Cr/Au sputtering and wet etching; (iii) transfer of the scanner pattern into the DL by DRIE using a thermal SiO_2_ hard mask; (iv) release in vapor HF; and, (v) laser-assisted bonding of glass microlens. The main difficulty that was observed during the fabrication of z-scanner is related to the processing of SOI wafer with through-hole in its HL. This wafer needs a backside protection to avoid liquid penetration while undergoing the wet processes (i.e., cleaning, photoresist development) or membrane collapse when the wafer is clamped on the vacuum chucks. The latter is of special attention while transferring the pattern on the suspended membrane during the photolithography. One of the critical fabrication steps is the dry etching of the suspended silicon platform by DRIE etch, which is very sensitive to thermal issues. In order to prevent any errors in the pattern transfer between the membrane and the cooled wafer body, the DRIE etch must be optimized to minimize the local overheating of the membrane/mask and hence to avoid the strong overetching of pattern. For this reason, only good quality, thermally stable mask layers, such as thermal SiO_2_, should be used.

The entire fabrication process of the z-scanner is schematically presented in Figure 16.

#### 3.2.3. Hybrid Integration of Glass Ball Microlenses

The bonding of discrete micro-optical components on the silicon microactuators at the wafer level is a challenge. The microlens has to be individually positioned and bonded to the fragile suspended structures of the microactuator wafer in a reliable way without causing stress-induced deformation while also maintaining unaffected optical performances (e.g., focal length, surface quality, etc.). If an intermediate bonding material is necessary, then its application on the bonding area has to be safe for the microactuator springs, whereas the thickness of the intermediate bonding layer has to be well controlled to ensure the required relative distance between two scanning microlenses. Moreover, the full compatibility of this intermediate bonding material with the process temperature of wafer-level anodic bonding is necessary. In this work, precise handling and positioning of the microlenses is realized by the use of a manual pick-and-place assembly station with a vacuum pick-up tool. The whole assembly procedure, which includes a pick-up of an individual microlens, transfer, positioning over the microactuator platform, and release of the vacuum tweezers, takes about 3–5 min. One of the difficulties is the electrostatic charging of the tweezers/microlens, which requires a grounding of the fine metal capillary tool. The assembly process is reversible—the microlens can be picked up from the microactuator platform without the risk of structure damage. Several methods of the hybrid integration of the glass ball microlens with the silicon microactuators have been experimentally performed:low temperature adhesive bonding,medium temperature bonding using intermediate glass frit material,high temperature direct thermal fusion of Si-glass (no intermediate materials), andlaser-assisted local thermal fusion of Si-glass (no intermediate materials).

Adhesive bonding with UV-curable optical glue is a simple and fast method to achieve the microlens assembly. However, the use of optical glue is not compatible with wafer-level assembly by anodic bonding, due the too high temperature of the latter (~350 °C) and the requirements for long-term reliability. Glass-frit bonding offers a medium process temperature and results in a very reliable heat-resistant assembly. However, glass frit paste (FX-11-0366, FERRO, Mayfield Heights, OH, USA) is difficult to be applied on the MEMS structure, because of its high viscosity, making the thickness control difficult. Thus, other methods, which do not require any intermediate material, are preferred. Direct thermal bonding in an atmospheric pressure furnace was investigated to obtain a sufficiently strong silicon-glass connection without the deformation of lens shape. Here, the stress generated during cooling, caused by the mismatch of coefficients of thermal expansion between N-BK7 glass (~8.3 × 10^−6^/°C) and silicon (~2.6 × 10^−6^/°C), had to be minimized. However, the silicon-glass fusion bond produces a slight deformation (tilt) of the entire platform, as well as deteriorating the microlens surface. Finally, a technique that is based on the laser-assisted bonding was selected for the assembly, because it offers a local and well-controlled melting of the glass material, which does not require an overall wafer heating. In this bonding scheme, the illuminating NIR light (λ = 808 nm) is mainly absorbed by silicon, generating the heat that is transferred to the silicon-glass interface. Although it allows well-localized bonding, it must be noted that the effective bonding requires large enough surface that is exposed to NIR radiation. Therefore, the glass microlens is much more difficult to bond to a highly structured Si platform via holes, as is the case of our *x-y* scanner, when compared to an unstructured platform. 

The glass ball microlens bonding was tested using the computer controlled laser station shown in Figure 17a. This is equipped with pigtailed high power NIR laser diode, collimating and focusing optics, imaging control system, and motorized *x-y* stage for sample positioning. The selected ball microlens was positioned in the center of a slightly focusing laser beam (Figure 17b), with a diameter of ~1 mm, followed by time or power-controlled laser exposure. Preliminary results, which were obtained with 1.3 W of laser power and a minimum exposure time of 30 s, indicate the successful bonding without any cracks at the Si-glass interface or deterioration of glass surface, as shown in Figure 17c. 

Figure 18 shows the final fabrication step of both microlens scanners, where the ball microlens is assembled on the movable platform of the scanner and then laser bonded, according to the described previously procedure.

It is worth mentioning that the proposed assembly method has an impact on the axial positioning of the microlens in the optical scanning system. The position of the microlens center above the wafer level (*h*) depends on the microlens radius (*R*) and assembly port radius (*r_p_*), as shown in Figure 19a. Figure 19b shows the positioning error (Δ*h*) that strongly depends on the assembly port radius for given microlens and assembly port diameter errors. For the assembly port radius *r_p_* = 200 µm, the positioning error is within the range of ±2.5 µm.

### 3.3. Assembly and Packaging of 3D Microlens Scanner

To vertically integrate all five heterogeneous building blocks of the 3D microscanner, as shown in Figure 20, a hierarchic assembly procedure has been proposed [12]. The functional blocks are assembled in a well-defined bonding sequence by using suitable bonding methods to simultaneously establish reliable mechanical and electrical connections. Finally, the hermetic sealing of the complete system is realized by carrying out the last bonding step under vacuum. Through this hierarchic assembly method, the five constitutive components of the 3D microscanner can be successfully joined together with a satisfactory bonding yield, which is estimated to be at least 30% in this stage of development. 

All the components of the 3D microscanners are organized in such a way that glass, LTCC, and silicon wafers are alternatively stacked. Through such an arrangement, all of the components of the final microsystem can be sequentially joined together by using several single anodic bonding processes. Anodic bonding was selected, since it is especially suitable to assemble the functional blocks into a complete optical system. This method allows a very strong and reliable connection between silicon and glass while using an intermediate temperature range (typically 350 °C), which do not harm the metallic thin-film layers. In addition, it does not need additional interface materials, which would increase the mechanical stress of the whole stack. However, the other materials that are used in the microsystem must be compatible. Therefore, the LTCC material, which was used for the spacer, has a similar CTE to the glass and Si and it is anodically bondable. Moreover, the average roughness of LTCC surface, as measured by AFM over 10 × 10 µm^2^, is Ra < 1 nm, hence, very close to that of glass.

The medium temperature of anodic bonding and the absence of bonding intermediate layers allow for improving the control of distance (gap) between the centers of two scanning ball microlenses. This gap, defined by optical design of scanning system (afocal arrangement) to be a sum of their focal lengths, should be respected within the tolerances below 5 µm. This is challenging due to several constraints that result from the wafer-based fabrication principle, for example, a wafer-to-wafer thickness dispersion of LTCC substrates can reach 20 µm, whereas the wafer bow can be in the wide range of 7–70 µm. Therefore, the LTCC substrates for spacer fabrication have been carefully selected for low thickness tolerance (±5 µm) and small wafer bow (<50 µm), being compatible with anodic bonding. 

The bonding strength of the Si-LTCC strongly depends on the temperature and the applied voltage [25]. In our tests, the bonding temperature has been changed between 300 °C and 400 °C, whereas the voltage was set to be 400 V or 800 V. It was found that, at 400 °C and 400 V, the bonding strength is comparable to that obtained when bonding Si to the borosilicate glass. It is worth noting that the Si-LTCC bonding strength decreases faster with bonding temperature than during the Si-glass bonding. However, additional plasma treatment can compensate this effect.

Figure 21 shows the bonding sequence. At the beginning of the bonding process a glass ball microlens has to be integrated into the z-scanner. Next, the wafer W4 (z-scanner) and wafer W3 (LTCC spacer) are aligned and bonded (Step 1). This first stack must be then bonded to the wafer W2 (*x-y* scanner, Step 2). In this case, the interface consists again of LTCC and Si layers. Afterwards, a second microlens has to be integrated into the *x-y* scanner. To close the system from the top, a top glass lid (W1) is anodically bonded to the scanner stack (Step 3). In the last step, a bottom glass lid W5 is bonded to the bottom side of the z-scanner to realize hermetic sealing (Step 4).

Figure 22 shows the cross-section of an exemplary multilayer stack fabricated by sequential anodic bonding. An inspection of bonding interface by scanning acoustic microscope, demonstrating a high bonding yield-rate > 90%. No crevice or delamination is observed along the whole length of the inspected interface. The bonded wafer stack was successfully diced into 5 mm size square chips. Both investigations indicate that reliable bonding has been established. 

In order to drive the electrostatic microactuators that are sandwiched between glass and ceramic layers, the through-wafer vias (TWV) were employed to create electrical connections from the pads, located on the top glass lid wafer (W1), through the stack of one glass wafer, two SOI wafers, and one ceramic wafer. Figure 23a shows the cross-sectional view of an individual chip. The vias on different levels of the stack are connected during sequential multi-level anodic bonding by using the pressure-contacts between Cr/Au (20/200 nm) pads. Tests of Au-Au contact formation have shown that a temperature at least 300 °C is necessary to form a low resistance ohmic contact. The thermo-compression bonding was done in a SUSS wafer bonder at 400 °C for 30 min at a tool pressure of 7 MPa. Before annealing, the electrical resistance between both of the wafers was in the range of 2.0–4.8 Ω, with a mean value of 3.5 Ω (contact area of 100 × 100 µm^2^) [26]. After an annealing step at 300 °C for 3 h, the contact resistance decreased to a mean value of 3.2 Ω, as shown in Figure 23b. The reliability of the contacts has been tested by thermal shock cycles (100 cycles, −40 °C–120 °C, 30 min–1 min–30 min). To compare the results, the pull strength was measured before and after the shock loading, as shown in Figure 23c. The results show that there is no obvious impact of bond time and shock loading to the fracture force of the gold/gold thermo-compression contact.

## 4. Conclusions

The lack of miniaturization in conventional 3D display and imaging systems limits their application fields and imaging capabilities. Responding to strong consumer demand for ultra-compact devices and global passion for “greener”, more power-efficient products, marketplace demands are also incorporating 3D integration into the mainstream. MOEMS technology that combines MEMS and micro-optics is well suited for the implementation of micro-optical scanners, because it enables the production of batch-fabricated microsystems at low cost. We demonstrated that the 3D packaging offers an effective integration platform for complex MOEMS. This packaging approach included the various heterogeneous technologies, structured in vertically stacked building blocks, including micro-optical a MEMS wafers. Thanks to the optimized design of such vertical architecture, the original assembly approach was investigated, offering the integrity of MEMS microactuators assembled with micro-optics and resulting in a low level of residual stress.

The present contribution includes the design, the fabrication process, and a fully integrated prototype of vertically integrated microlens scanner, which are suitable for a wide number of imaging systems and particularly for the on-chip confocal microscopy. The proposed 3D microlens scanner includes series of vertically stacked building blocks, combining the beam splitting functions with the 3D transmissive scanning. The presented results experimentally demonstrate the proof-of-concept of our approach. The first challenge of the proposed microsystem resides in the appropriate 3D packaging that combines several dies vertically by using multi-wafer bonding. The second challenge is to optimize the design and find the best solution for heterogeneous integration technologies to combine micro-optics and MEMS, which allows for high-frequency scanning and precision focusing of optical beams. The proposed approach of miniaturization of such a microsystem and the development of miniaturized opto-mechanical components that comprise our microsystem, made with a simplified optical design and minimizing the number of the components, permits the configuration with an optimal optical performance. The architecture of a microsystem constructed by MEMS micromachining is strongly dependent on the constraints of its fabrication process. One of the originalities of this work is the development of an adapted design platform, where the 3D assembly of different functional wafers is made by multi-wafer bonding and the optical performances (high resolution, control of aberrations) are conserved. This contribution also discussed the successful bonding of the silicon MEMS microactuators (*x-y* scanner, z-scanner, including fabrication parts), with other glass and ceramic components of 3D micro scanner, as well as their fabrication and simultaneous realization of the electrical connections by vertical through-wafer vias.

In future work, the optical performances will be improved by a replacing the ball microlenses by plano-convex microlenses, with improved optical quality being obtained per example by the deposition of antireflection coatings on optical surfaces.

## Figures and Tables

**Figure 1 micromachines-10-00185-f001:**
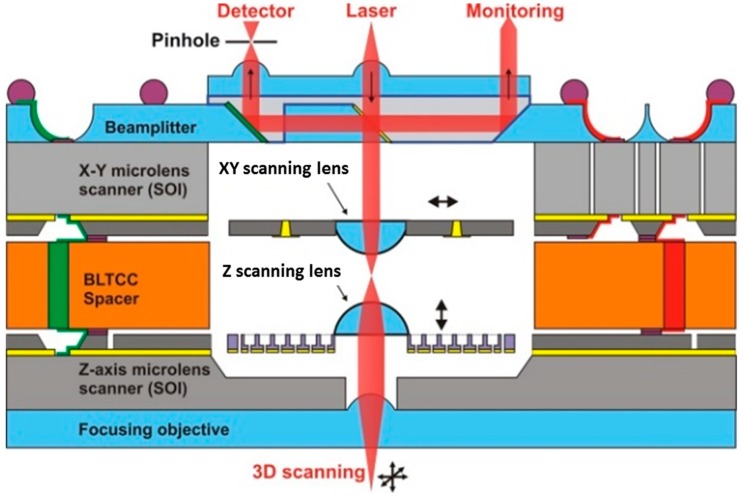
Schematic of the three-dimensional (3D) microscanner for on-chip scanning confocal microscope. Reproduced with permission from [9], published by SPIE, 2013.

**Figure 2 micromachines-10-00185-f002:**
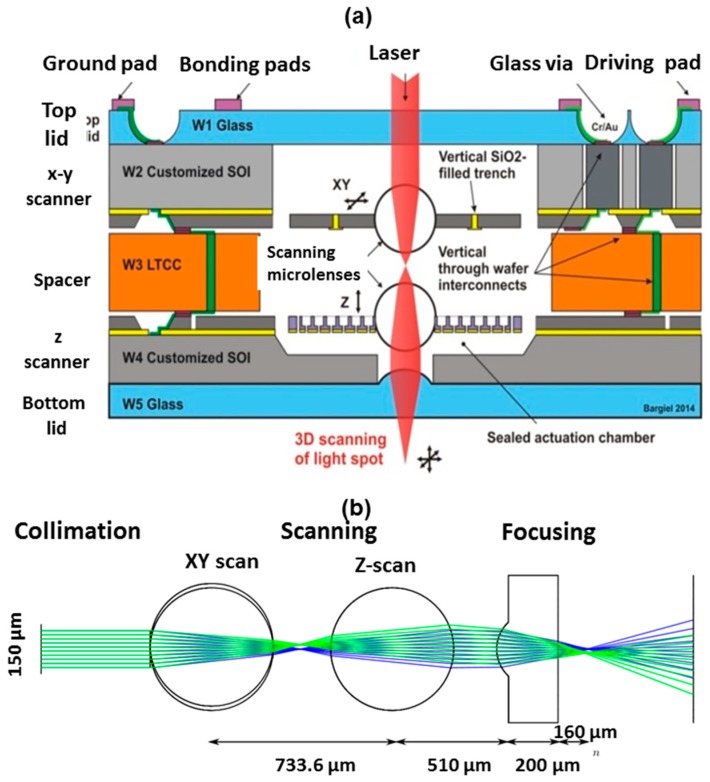
3D microoptical scanner: (**a**) cross-sectional view of microscanner; and (**b**) optical design of the microscanner based on three blocks: collimation block, scanning block with afocal doublet, and focusing block.

**Figure 3 micromachines-10-00185-f003:**
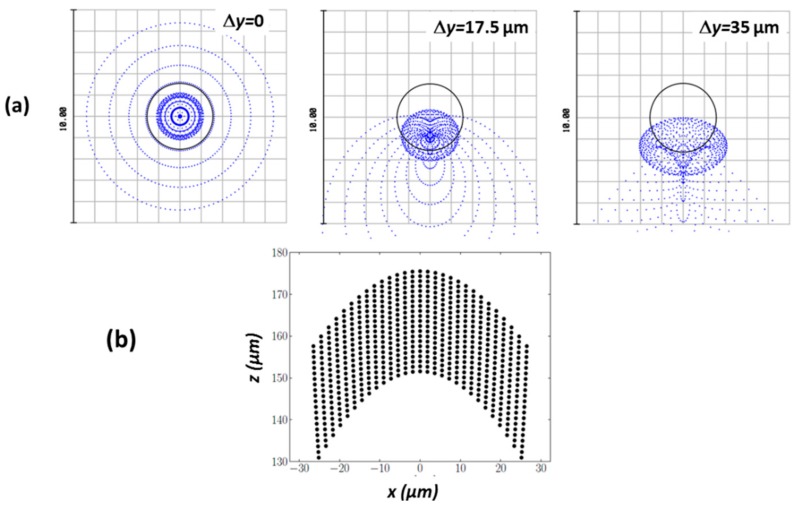
Optical performance of glass-ball microscanner: (**a**) spot diagrams for different lateral displacements of Δ*y* = 0 µm, 17.5 µm, 35 µm; and (**b**) shape of scanning volume (adjacent points correspond to the scanning lenses displaced by 2 µm in two considered (*x*,*z*) directions). Reproduced with permission from [13], published by OSA Publishing, 2015.

**Figure 4 micromachines-10-00185-f004:**
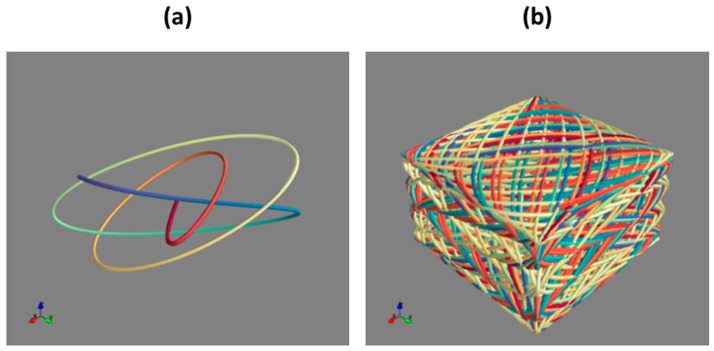
3D Lissajous curve obtained for the oscillation frequencies at *f*_x_ = 650 Hz and *f*_z_ = 452 Hz with different scanning times: (**a**) t = 5 ms; and, (**b**) t = 200 ms (color of the curve corresponds to the time moment).

**Figure 5 micromachines-10-00185-f005:**
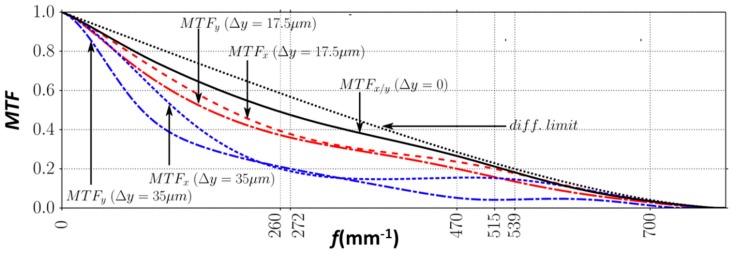
Modulation transfer function plotted for three situations of scan: lenses aligned on the optical axis (Δ*y* = 0, black solid curve), lens scan Δ*y* = 17.5 μm (red curves), and Δ*y* = 35 μm (blue curves). MTFy and MTFx represent the MTF cross-sections along and perpendicular to the direction of the lens displacement, respectively. Reproduced with permission from [13], published by OSA Publishing, 2015.

**Figure 6 micromachines-10-00185-f006:**
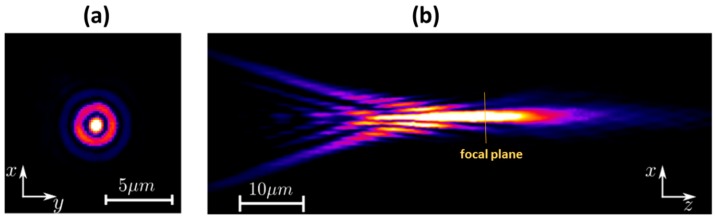
Measured Intensity Point Spread Function (IPSF) of micro ball lens (500-μm diameter), measurement in transmission at NA = 0.45: (**a**) image of the focal plane (XY slice) that was obtained at the focal plane; and, (**b**) XZ slice of IPSF along the optical axis.

**Figure 7 micromachines-10-00185-f007:**
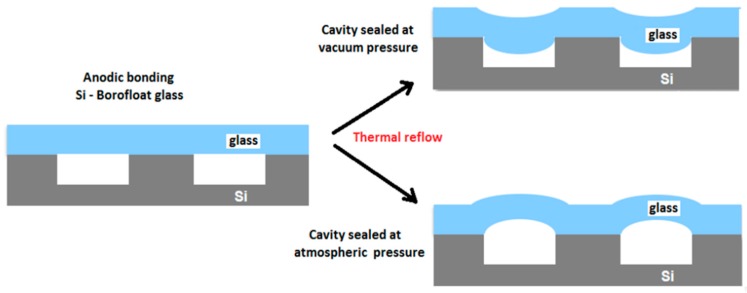
Principle of glass micro-blowing fabrication process. Reproduced with permission from [18], published by SPIE, 2018.

**Figure 8 micromachines-10-00185-f008:**
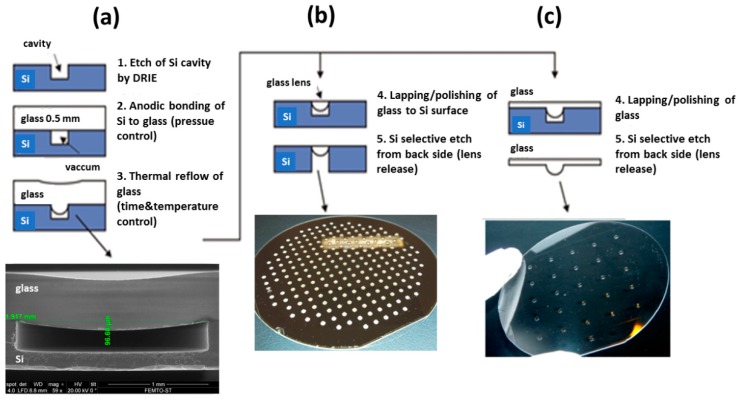
Wafer-level process of plano-convex glass microlenses: (**a**) reflow of glass over cylindrical silicon cavity, (**b**) planarization of glass surface and releasing of microlens by localized back-side Si dry etching or (**c**) by complete removing of silicon substrate. Reproduced with permission from [19], published by SPIE, 2015.

**Figure 9 micromachines-10-00185-f009:**
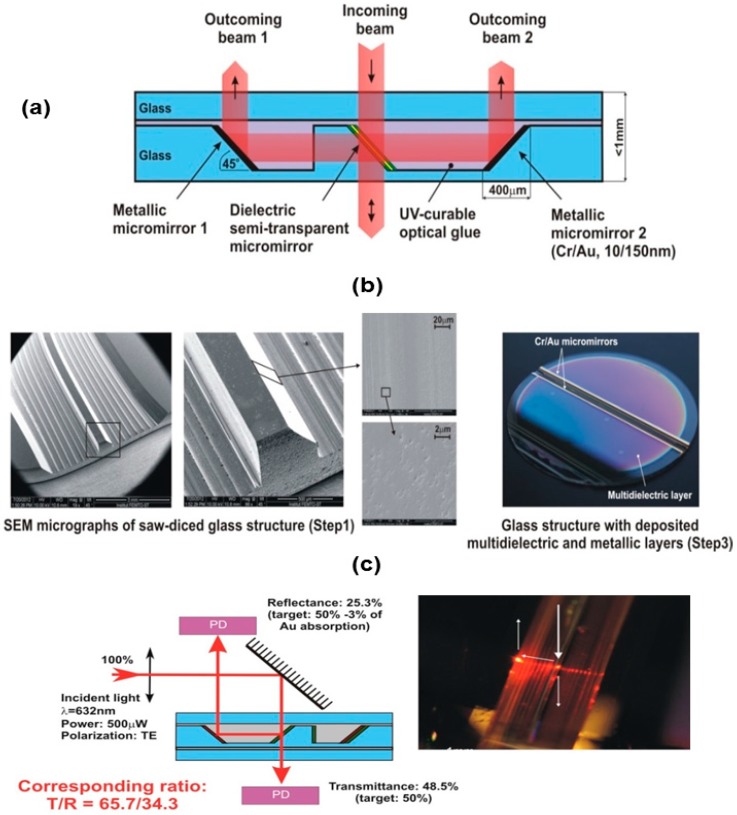
Wafer-level fabricated micro beam-splitter: (**a**) schematic view of the component, (**b**) use of customized blades to obtain inclined optical surfaces and SEM of 400-µm deep saw diced facets; and, (**c**) results of optical characterization (PD—photodiode).

**Figure 10 micromachines-10-00185-f010:**
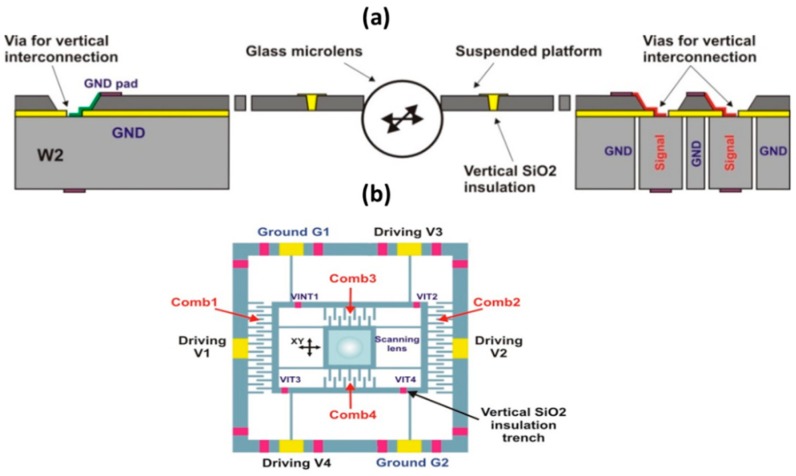
Electrostatic comb-drive *x*-*y* microlens scanner: (**a**) construction; and, (**b**) schematic of frame-in-the-frame architecture. Reproduced with permission from [11], published by SPIE, 2009.

**Figure 11 micromachines-10-00185-f011:**
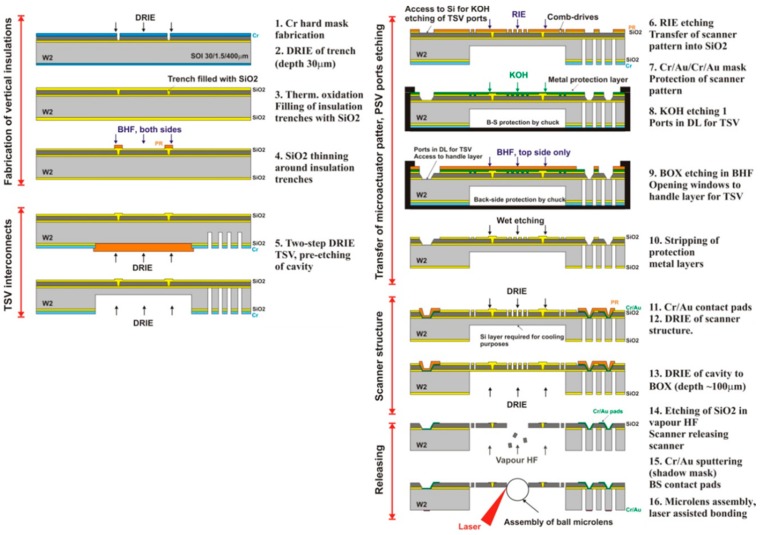
Flow-chart of the *x-y* scanner.

**Figure 12 micromachines-10-00185-f012:**
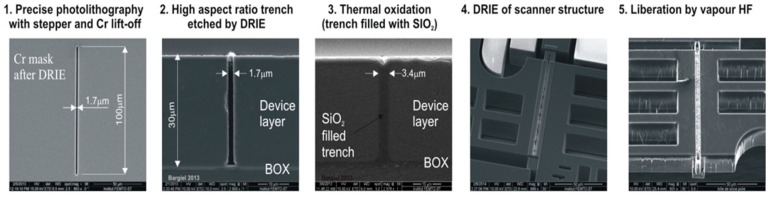
Fabrication steps of the vertical insulation trenches.

**Figure 13 micromachines-10-00185-f013:**
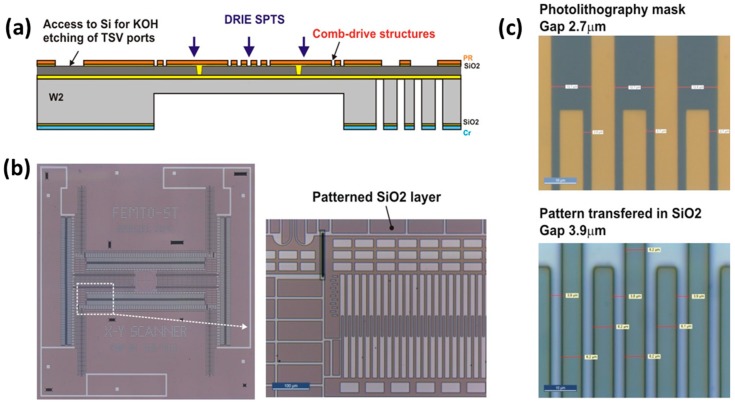
Transfer of precise microactuators pattern into the SiO_2_ hard mask layer by use of inductively coupled plasma (ICP) deep reactive ion *etching* (DRIE): (**a**) the flow chart, (**b**) patterned SiO_2_ layer; and, (**c**) measurement results of comb-drive fingers.

**Figure 14 micromachines-10-00185-f014:**
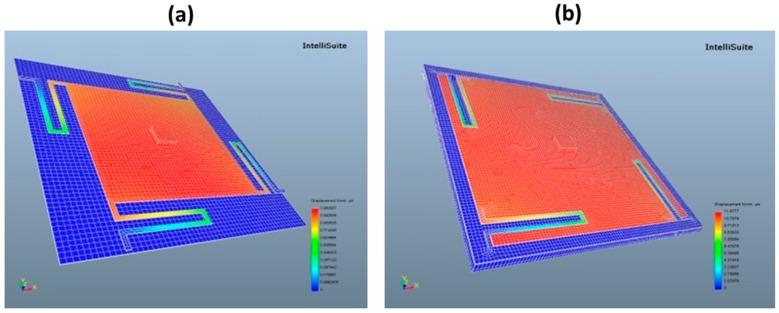
Electrostatic z-axis scanner: (**a**) construction; and, (**b**) modeling results (Intellisuite)—first resonance mode (piston) of spring suspensions.

**Figure 15 micromachines-10-00185-f015:**
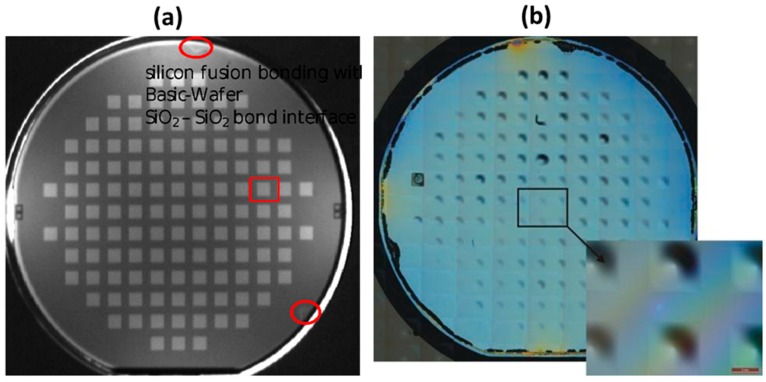
Customized “home-made” silicon-on-insulator (SOI) substrate with embedded cavity for z-scanner: (**a**) *Infrared* (IR) transmission of bonding interface; and, (**b**) microscope image of the membrane side.

**Figure 16 micromachines-10-00185-f016:**
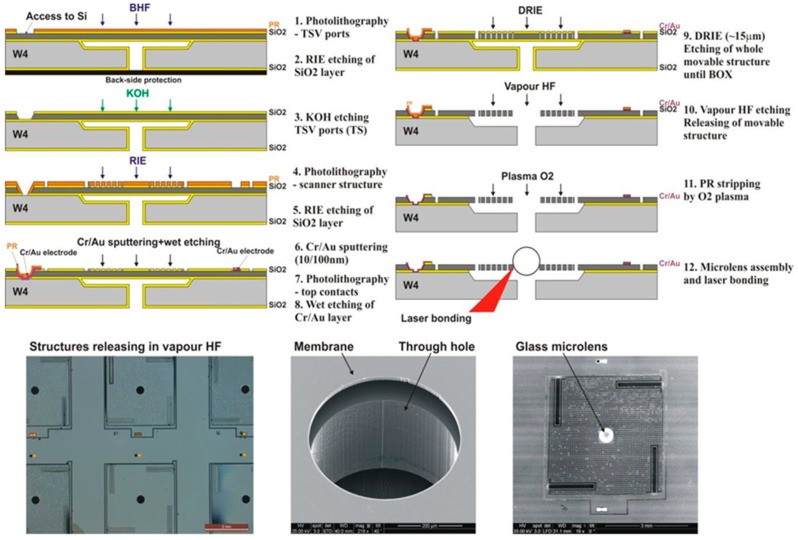
Flow chart of technological process of z-scanner.

**Figure 17 micromachines-10-00185-f017:**
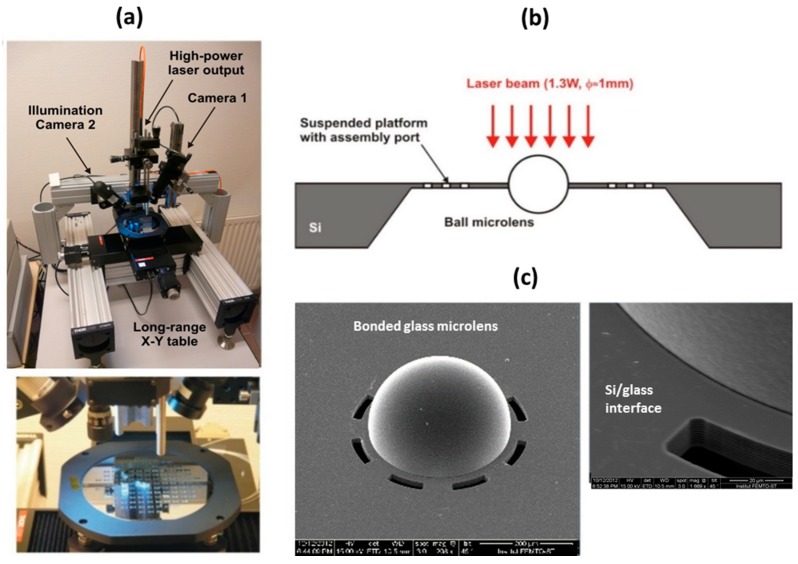
Laser-assisted bonding of glass ball microlens with silicon membrane: (**a**) high-power laser bonding station; (**b**) schematic drawing of the process; and, (**c**) SEM micrograph of bonded glass microlens.

**Figure 18 micromachines-10-00185-f018:**
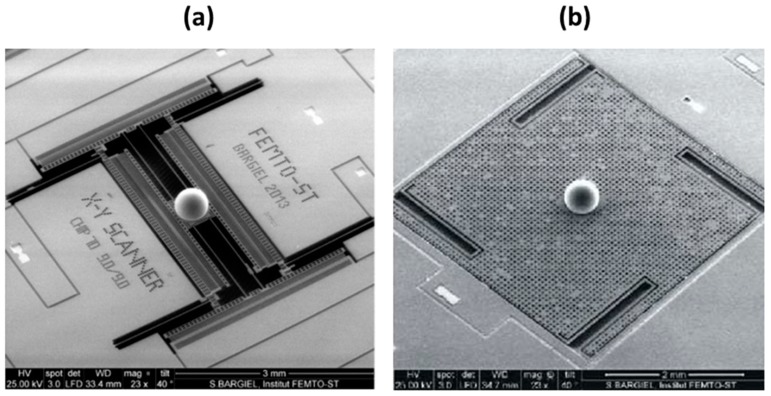
Fabricated microlens scanners after microlens integration: (**a**) SEM of individual chip of *x-y* microscanner and, (**b**) SEM of individual chip of z microscanner. Reproduced with permission from [13], published by OSA Publishing, 2015.

**Figure 19 micromachines-10-00185-f019:**
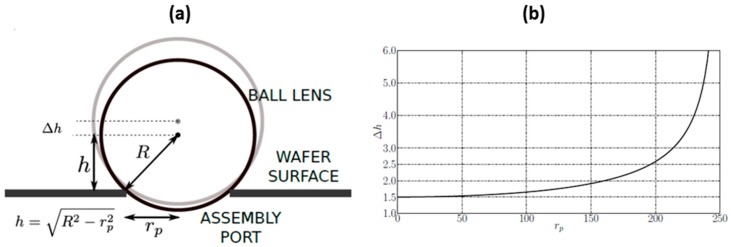
Assembly of ball microlens: (**a**) positioning of the ball microlens (2*R* = 500 µm) into a circular assembly port of radius *r_p_*; and, (**b**) error in axial positioning of the microlens strongly depends on assembly port size (plot of error assuming microlens radius deviation as 1.5 µm and port diameter error of 1.0 µm).

**Figure 20 micromachines-10-00185-f020:**
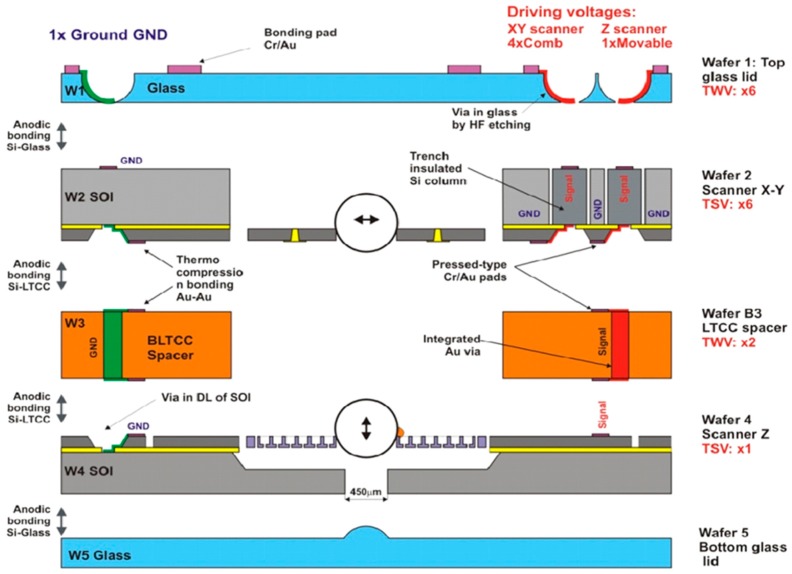
Details of the vertical integration strategy—cross-sectional view of 3D microscanner.

**Figure 21 micromachines-10-00185-f021:**
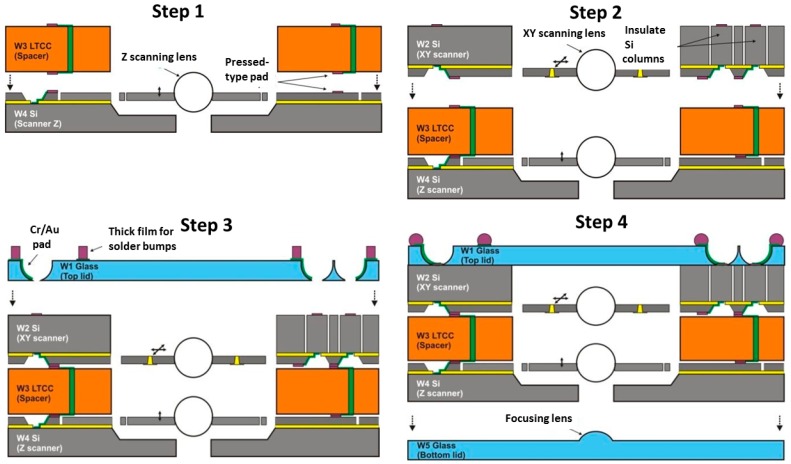
Assembly of 3D microscanner by sequence of anodic bonding processes.

**Figure 22 micromachines-10-00185-f022:**
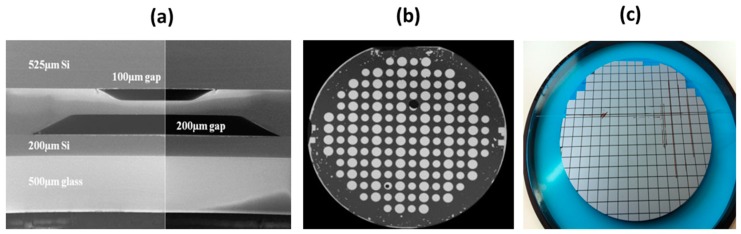
Multi-wafer test anodic bonding for 3D microscanner: (**a**) cross-section of anodically bonded Si-Glass-Si-Glass stack of structurized wafers; (**b**) inspection of bonding quality using scanning acoustic microscopy; and, (**c**) successful dicing experiment.

**Figure 23 micromachines-10-00185-f023:**
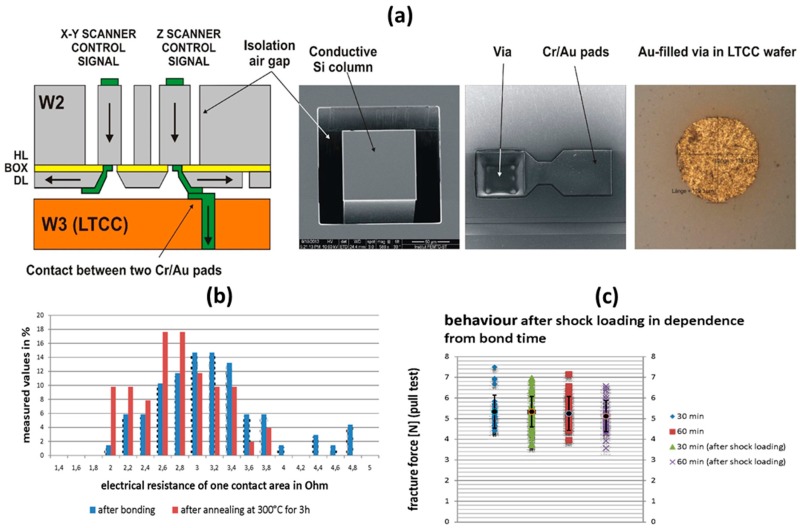
Electrical interconnection for individual microactuators, based on air-insulated conductive Si column, Cr/Au contact pads and Au-filled vias embedded in low temperature co-fired ceramic (LTCC): (**a**) details; (**b**) influence of annealing step on the electrical resistance of contact; and, (**c**) distribution of fracture force (pull test) before and after shock loading as a function of bonding time (30 min and 60 min).

**Table 1 micromachines-10-00185-t001:** Expected optical performances of glass-ball microscanner.

Parameter	
Numerical aperture	0.45
Volume of scanning volume, low aberration-level (lateral displacements Δ*xy* = ±35 μm; Δ*z* = ±20 μm)	60 × 60 × 25 µm^3^
Lateral resolution	1.9–3.8 µm
Axial resolution	13 µm

**Table 2 micromachines-10-00185-t002:** Expected performances of the *x-y* scanner.

Parameter	Value	
	*x*-axis	*y*-axis
Resonance frequency (with 500-µm microlens)	500 Hz @ 8 µm spring	590 Hz @ 8 µm spring
	700 Hz @ 10 µm spring	830 Hz @ 10 µm spring
Displacement at resonance	min. ± 35 µm @ < 60 V	
Quality factor	~50	~70

**Table 3 micromachines-10-00185-t003:** Expected parameters of z-scanner.

Parameter	Value
Resonance frequency (with 500-µm microlens)	570–600 Hz
Displacement *x*-axis at resonance	min. ± 20 µm @ < 70 V
Static critical voltage	88 V
Dynamic critical voltage	82 V

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
