# Peer review of "Technological Platform for Vertical Multi-Wafer Integration of Microscanners and Micro-Optical Components"

_micromachines, 2019, doi:10.3390/mi10030185_

Round 1
Reviewer 1 Report
Please see the attached file.

Author Response
The response to the revviewer is included here as a word file

Reviewer 2 Report
This manuscript described a complete work of integrated multi-layer MEMS-based confocal microscope. The microfabrication process flow was illustrated in detail with solid data support. Some minor aspects would expect authors to address in order to further polish the content of the manuscript.
1. In section 2, authors should put more efforts on the optical performance of the proposed system. For example, in Figure 3, It is very difficult to understand Figure3. The resolution of all the sub-figures are poor. Labeling of the axes should be clearly presented and explained. Figure3(b) requires more explanation how to form the scanning volume.
2. In Figure 4, What is the optical focusing performance with respect to the resonance driving trajectory?
3. In Figure 5, it would be much better to provide the real optical focusing performance of the actual system. The shown single ball lens performance is not sufficient to characterize the overall system performance.
In general, I would recommend this manuscript to be accepted as minor revision.
Author Response

(The authors gave the same response as above.)

Reviewer 3 Report
Dear Editor and Authors,
I have read the paper entitled "Technological platform for vertical multi-wafer integration of microscanners and micro-optical components for on-chip microscopy", which deals with a very interesting and relevant topic that is attractive for many readers interested in novel/modern MEMS/MOEMS devices and/or micromachining processes. Initially it seemed it would be a welcome change to finally have an easy-to-read paper written in fluent and proper English. However, a huge amount of minor corrections related to proper English/formulation are required (a clear sign -in my opinion- that they have proof-read it among themselves and correct it thoroughly before submission and just rushed to submit it to the Journal as quickly as they could, a rather callous attitude if I may say). Fortunately, content-wise I cannot raise any significant objections. It is basically a paper summarizing the new successful integration of a very complex optical microsystem, which is already quite detailed and long (though relatively well-written, if one makes exception of the annoyingly numerous minor English errors). Since this is not a typical investigative research report, I do not think one can apply here the usual standard evaluation questions & criteria, and in its special/specific class I believe it is a very good paper.
Given the need to make the enormous amount of English corrections (see below), my final recommendation is that MINOR corrections are needed.
The corrections/additions the paper needs are listed below, as follows:
MAJOR:
N.A.
MINOR:
- The minor but very many required English corrections are:
In line 25 of p.1, replace "offers" (the very last word) with "offer".
In line 39 of p. 1 replace "assembly" with "assemble".
In line 41, I believe it may be better to replace "propose technological issues of vertically integrated" with "propose solutions to the technological issues raised by the vertical integration of".
In line 51 at top of p.2, replace "connection" with 'connections".
In line 72 at p.3 (3rd line from the top), delete "the schema of"
In line 84 at p.3, replace "incoming laser beam" with "incoming illumination (laser beam)"
In line 86 at p.3, replace "environmentally induced" with "environmentally-induced". Also, personally I would start a new paragraph with the next following sentence.
In line 89 at p.3, replace "to matched coefficient of thermal expansion" with "to their matched coefficients of thermal expansion (CTEs).". Later in the same line but in the next sentence, replace "To drive" with "To drive the".
In line 90 at p.3 delete the commas ",".
In line 98 at p.3, replace "of focused" with "of a focused".
In line 99 at p.3, replace "of glass ball NA = 0.45" with "of the glass ball (NA = 0.45)".
In line 101 at p.3, replace "limit optical performances of microscanner" with "limit the optical performances of the microscanner".
In line 118 at p.4, replace "finite" with "the finite", and later in the same sentence, in line 119, replace "and still working" with "while still allowing operation".
In lines 120-121 at p.4, replace "Exemplary 3D Lissajou+s patterns, presented in Fig. 4," with "The examples of 3D Lissajous patterns presented in Fig. 4", and later in the same sentenece & line, replace "frequencies of scanning" with "scanning frequencies".
In line 138 at p.5, replace "is that all-resonance scanning" with "of all-resonance scanning is that it".
In line 146 at p.5, replace "microscanner employs glass ball microlenses from" with "the microscanner employed glass ball microlenses made of", and in the same sentence on the next line, replace "are" by "were".
In line 148 at p.5, replace "of microlenses" with "of these microlenses".
In line 149 at p.5, replace "The principle of measurements consists on sequential recording of lateral slices the 3D" with "The measurements consisted of sequentially recording the images of lateral slices of the scanned 3D", and later in the same sentence -but on the next line- replace "by investigated" with "by the investigated".
In line 150 at p.5, replace "measured IPSF of" with "the measured IPSF of a"
In line 153 at p.5, replace "at left" with "in the left side of the image". Also, when the image shown in Fig.5(a) is referred to in the text -or maybe better in the caption of Fig.5 itself- the Authors should specify to which z location does that cross-sectional image of the ISPF corresponds.
In line 160 at p.5, replace "assembly of ball-microlens" with "ball-microlens assembly".
In line 163 at p.6, replace "borosilicate glass" with "in borosilicate glass the", and in the last part of the same sentence, but on the next line, replace "propose the technique of" with "used micro-scale"
In line 173 at p.6, replace "for" with "necessary for".
In line 174 at p.6, replace "plano-convex microlens" with "the plano-convex microlenses".
In line 176 at p.6, replace "in" with "in a" and then delete "the" before "DRIE".
In line 178 at p.6, replace "to" with "to the".
In line 179 at p.6, replace "the atmospheric furnace" with "a furnace at normal atmospheric pressure".
In line 180 at p.6, replace "cavity" with "the cavity".
In line 181 at p.6, replace "temperature" with "the high temperature", and later in the same sentence replace "lens" with "a lens".
In line 181 at p.6, replace "become" with "becomes".
In line 187 at p.6, replace "because" with "because it is a".
In line 192 at p.7, replace "ball microlens system" with "a ball microlens".
In line 196 at p.7, replace "Additional" with "An additional".
In line 197 at p.7, replace "in case of" with "for the"
In line 200 at p.7, replace "3D microlens scanner in" with "a 3D microlens scanner into a".
In line 212 at p.8, replace "the wafer-" with "wafer-"
In line 214 at p.8, replace "beam" with "beams".
In line 215 at p.8, replace "use of" with "using an"
In line 219 at p.8, replace "facing mirror to the" with "mirror facing the".
In line 220 at p.8, replace "incident" with "the normally incident light beam", and later in the same sentence replace "TE-polarized light, 48.5%" with "48.5% TE-polarized light".
In line 224 at p.8, replace "works need to focus on decreasing of optical losses by further improvement of" with "work needs to focus on decreasing the optical losses by further improving the".
In line 225 at p.8, replace "Complete integration of the micro-beam splitter with" with "The complete integration of the micro-beam splitter with the".
In line 227 at p.8, delete the comma "," after "component"; later in the same sentence replace "within wafer level process" with "with a standard process at wafer level", and further in the same sentence but in the next line replace "on" with "at".
In line 230 at p.8, replace "scanning MEMS actuators," with "the scanning MEMS actuators and the"
In line 235 at p.9, replace "the fabrication" with "fabrication", and further on replace "of x-y" with "of the x-y" (Again, make "x-y" & "z" italic in the rest of the places where they are located in this paragraph).
In lines 236-237 at p.9, replace "z-scanner and adapted for the carrying" with "a z-axis scanner and adapted to also act as a supporting holder".
In line 241 at p.9, replace "x-axis" with "the x-axis"
In line 243 at p.9, replace "; attached to the inner frame, displace only the platform along y-axis" with "attached to the inner frame, displace the platform only along the y-axis".
In line 244 at p.9, replace "two lateral directions x-y of actuation" with "the two lateral directions of actuation (x or y)", and later on in the same setentence -but in the next line- add "the" before "required". Similarly, in line 248, add "the" before "x-y" (as is correctly done in the caption of Table 2!).
In line 254 at p.10, replace "resulting structures are" with "the resulting structures are also". In the next sentence, add "a" in several places, namely: before "4'' ", before "30", before "400" and before "1.5", respectively.
In line 257 at p.10, replace "required to process the wafer" with "processed", later in the same sentence replace "all structures of" with "the structures of all" while also deleting (in line 258) "of SOI".
In line 259 at p.10, replace "Vertical" with "The vertical", and later in the same sentence -but on the next line- replace "connection of" with "connections to the".
In line 267 at p.10, replace "are performed using" with "was performed using the", and in the next sentence on the same line, replace "use the" with "used"
In line 268 at p.10, replace "BOX" with "the BOX" and replace "of" with "of the".
In line 269 at p.10, replace "to" with "to the"
In line 270 at p.10, replace "steps of fabrication" with "fabrication steps".
In line 272 at p.10, replace "DL by" with "the DL by an".
In line 273 at p.10, replace "under etching of" with "underetching of the" .
In line 275 at p.10, replace "of" with "of the" and later in the same sentence replace "in ~2.1 μm" with "in a ~2.1 μm thick".
In line 282 at p.11, replace "consists, on" with "consists of".
In line 284 at p.11, replace "suitable dry SiO2 etching method, providing minimal under etching of" with "a suitable dry SiO2 etching method that provides minimal underetching of the".
In line 285 at p.11, replace "of" with "of the", then later in the same sentence replace "into 1.6 μm SiO2 layer based on" with "into the 1.6 μm SiO2 layer by".
In line 286 at p.11, replace "in the" with "using a".
In line 287 at p.11, replace "in" with "in the", and later in the same sentence both at the end of this line and the beginning of the next one, replace "under etching" with "underetch".
In line 288 at p.11, replace "of BOX" with "the BOX layer".
In line 289 at p.11, replace "of ball" with "of the ball".
In line 294 at p.12, replace "of" with "of the", and later in the same sentence replace "on" with "on the".
In line 296 at p.12, replace "Movable" with "The movable", and later in the same sentence replace "by DRIE of device layer of" with "after DRIE of the device layer of the".
In line 297 at p.12, replace "movable electrode of" with "the movable electrode of the", and later in the same sentence replace the very last word "of" with "of the".
In lines 298 & 299 at p.12, replace "counter electrode" with "counterelectrode"
In line 298 at p.12, delete "use of".
In line 302 at p.12, replace the very last word "of" with "the".
In line 304 at p.12, replace "of" with "of the".
In line 307 at p.12, replace "Fabrication of z-scanner requires "home-made" customized SOI substrate with" with "The fabrication of the z-scanner requires a "home-made" customized SOI substrate with a".
In line 308 at p.12, replace "over embedded" with "suspended over an embedded", and later in the same sentence replace "through hole" with "through-hole".
In line 310 at p.12, replace "thermal" with "a thermal".
In line 310 at p.12, replace "of" with "of successive cleaning steps using the standard solutions".
In line 316 at p.12, replace "recorded by" with "monitored by a."
In line 317 at p.12, replace "further" with "subsequent".
In line 319 at p.12, replace "through holes" with "through-holes", and at the very end of the sentence delete "method".
In line 320 at p.12, replace "Optical" with "An optical", and later in the same sentence replace "the SOI" with "the resulting final".
In line 325 at p.13, replace "The process" with "The subsequent process necessary to finalize the fabrication of the entire system", and later in the same sentence replace "device layer based on the" with "the DL using".
In line 327 at p.13, replace "device layer by DRIE method by use of" with "the DL by DRIE using a".
In line 328 at p.13, replace "releasing process" with "release", and later in the same line but in the next sentence, replace "Main" with "The main".
In line 329 at p.13, replace "of" with "of the" in both places in that sentence, and the end of that line & the beginning of the next one, replace "through hole" with "through-hole".
In line 330 at p.13, replace "Such" with "This", and later in the same sentence replace "during" with "while undergoing".
In line 331 at p.13, replace "when" with "when the".
In line 332 at p.13, replace "transfer" with "transfering".
In line 333 at p.13, delete "process" and later in the same sentence replace "steps of fabrication" with "fabrication steps".
In line 334 at p.13, replace "consists on" with "is" and delete "method".
In line 335 at p.13, replace "Due to bad thermal transfer between" with "In order to prevent any errors in the pattern transfer between the"
In line 336 at p.13, replace "DRIE" with "the DRIE"
In line 337 at p.13, replace "over etching" with "overetching".
In line 338 at p.13 add a comma "," after "SiO2".
In line 339 at p.13, replace "Technological process" with "The entire fabrication process".
In line 343 at p.13, replace "fragile" with "the fragile".
In line 344 at p.13, replace "of" with "of the".
In line 346 at p.14, replace "keeping" with "maintaining unaffected"
In line 347 at p.14, replace "for" with "for the".
In line 350 at p.14, replace "with" with "with the", and later in the same sentence replace "of" with "of the".
In line 352 at p.14, replace "Whole" with "The whole".
In line 354 at p.14, replace "platform and releasing" with "the platform and release".
In line 355 at p.14, replace "of" with "of the".
In line 356 at p.14, replace "from" with "from the".
In line 357 at p.14, replace "of glass" with "of the glass".
In line 363 at p.14, delete "method".
In line 365 at p.14, replace "due to too" with "due to the latter's too"
In line 366 at p.14, delete "method" and later in the same sentence replace "and well" with "and results in a very".
In line 368 at p.14 add a comma "," after "viscosity".
In line 369 at p.14, replace "intermediate" with "any intermediate".
In line 370 at p.14, replace "atmosphere" with "atmospheric pressure", and later in the same sentence replace "sufficient" with "sufficiently strong".
In line 371 at p.14, replace "be" with "by the".
In line 373 at p.14, replace "Silicon-glass fusion bond" with "However, the silicon-glass fusion".
In line 374 at p.14, replace "generates a deterioration of" with "deteriorates the".
In line 375 at p.14, replace "of" with "of the".
In line 377 at p.14 add a comma "," after "silicon" and later in the same sentence replace "transfer" with "transferred".
In line 379 at p.14, replace "requires" with "requires a" and later in the same line but in the next sentence replace "Glass microlens is therefore much" with "The glass microlens is, therefore, much more".
In line 380 at p.14, replace "for" with "is the case for our".
In line 381 at p.14, replace "a no-structured" with "an unstructured" and later in the same line but in the next sentence replace "A test of glass ball bonding was performed" with "The glass ball bonding was tested".
In line 387 at p.14, replace "on" with "at the".
In line 393 at p.15, replace "step of fabrication" with "fabrication step", and add a comma "," after "scanners".
In line 395 at p.15, replace "here" with "previously".
In line 399 at p.15, replace "vertically" with "vertically all".
In line 400 at p.15, replace "Fig. 18," with "Fig. 18, a",
In line 401 at p.15, replace "in" with "in a" and later in the same sentence replace "use" with "using".
In line 403 at p.15, delete "condition".
In line 405 at p.15, replace "with" with "with a" and delete "-rate".
In line 408 at p.16, replace "appear" with "are stacked" and later in the same line, nbut in the next sentence, replace "such" with "such arrangement, components" with "such an arrangement, all the components of the final microsystem".
In line 409 at p.16, replace "The anodic" with "Anodic".
In line 410 at p.16, replace "fitted" with "suitable".
In line 411 at p.16, replace "at the medium" with "using an intermediate".
In line 414 at p.16, replace "is applied for" with "was used for the".
In line 417 at p.16, replace "the tests, the bond" with "our tests the bonding", and later add a comma "," after "400°C".
In line 419 at p.16, replace "to borosilicate" with either "to that obtained when bonding Si to borosilicate", OR with "to the intrinsic mechanical resistance of bulk". Later, in the same line, but in the following sentence, replace "to note" with "noting".
In line 430 at p.17, replace "cross section of" with "cross-section of an".
In line 431 at p.17, delete "by" and later in the same sentence replace "shown" with "shown a".
In line 432 at p.17, replace "is" with "was".
In line 433 at p.17, replace "indicate" with "indicated".
In line 434 at p.17, replace "drive" with "drive the", and delete the comma "," after "microactuators".
In line 435 at p.17, replace "technology of through wafer vias (TWV) is applied" with "through-wafer vias (TWV) were employed".
In line 437 at p.17, replace "of" with "of an".
In line 442 at p.17, replace "use of" with "using".
In line 461 at p.18, delete "of".
In line 462 at p.18, replace "offering" with "it enables the production of", and later in the same line, but in the next sentence, replace "demonstrates that the" with "demonstrated that".
In line 463 at p.18, replace "offers" with "offers an effective" and later in the same sentence delete "the effective integration of".
In line 464 at p.18, replace "disposed" with "structured".
In line 466 at p.18, delete the comma "," after "technologies".
In line 467 at p.18, replace "offering" with "resulting in".
In line 467 at p.18, replace "technology with "fabrication process".
In line 471 at p.18, replace "of" with "of a".
In line 472 at p.18, delete "the".
In line 473 at p.18, delete "series of", and later on in the same sentence replace "blocks, combining" woth "blocks that combine".
In line 474 at p.18, replace "with the" with "with".
In line 475 at p.18, replace "of proposed" with "of the proposed".
In line 479 at p.18, replace "such" with "such a".
In line 480 at p.18, replace "of" with "of the".
In line 481 at p.18, add a comma "," after "components".
In line 482 at p.18, replace "making optimal the" with "with an optimal".
In line 483 at p.18, replace "process flow" with "its fabrication process".
In line 485 at p.19, replace "discuss also" with also discussed".
In line 486 at p.19, replace "methods of" with "successful".
In line 487 at p.19, replace "of" with "of the".
In line 489 at p.19, replace "and simultaneous" with "while simultaneously realizing" and later in the same sentence replace "through wafer technology" with "through-wafer vias".
In line 490 at p.19, replace "works" with "work" and delete "a modification of the".
In line 491 at p.19, replace "replacement of ball microlenses by" with "replacing the ball microlenses with".
In line 492 at p.19, replace "per example" with ", e.g.," and delete "the".
- The sizes of the drawings in Fig.1 & 2 are too small, one can barely read the text in the different parts of the drawing. I suggest the Authors increase significantly the size of these drawings. Also, indicate the reference (is it [6]?) in the caption of Fig.1 as well. For Fig.2, to make larger and more visible both drawingas (a) & (b), I suggest that the Authors stack them vertically instead of side-by-side.
- The x-y microactuator has been referred to a previous realization (ref. [7]); has the z microactuator also been described in Literature previously? f yes, please mention the reference. Also, make "x-y" and "z" italic.
Furthermore, the Authors should specify the voltages necessary to control the x-y & z microactuators, respectively. (If not the full range, then at least the max. values required for max. deflection).
- In lines 87-88 at p.3, the Authors state the necessity of assembly with tolerances of a few micrometers. However, the way they phrased the next sentence ("This is achieved...") is made in such a way which -in my opinion- seems to highlight the specific usage of the LTCC. However, again in my opinion, the choice of the spacer material is only a part of the success of achieveing the desired tolerances. How did the Authors achieve a very good planarity (what value specifically?), and especially surface roughness (what value specifically?) for the materials to be bonded, and how was the bonding optimized & performed to maintain the planarity and ensure the desired accuracy for the gap space? I believe the Authors should provide these answers in the paper.
- The caption of Fig.3 should be brought on p.3, right under the Fig. itself.
- Shift at the top of the page (next page, p.6) the title of subsection 3.1.2.
- Indicate in the caption of Fig.8(c) that the "PD" shown in the drawing means "photodiode" for the read-out of the desired optical signals.
- At p.8 the Authors discuss the dicing procedure and state that it "was investigated in terms of optimal dicing parameters and cleaning procedure of the blade". So which are the values of the optimal dicing recipe's parameters?
- Bring all the parts of Fig.9 on the same page.
- Shift section 3.2.2 and the subsequent Fig.13 to be fully contained onto the next page (page 12, if the page numbering of the revised manuscript will be identical with the current one).
- Make sure that Fig.14 is fully contained onto page 13 (right now the "a)" and "b)" are at the bottom of p.12).
- Increase the size of Fig.15 to occuppy the entire width of the page. This will also shift onto the next page Section 3.2.3, whose first lines are at the bottom of p.13.
- At p.16, take the three lines of text (393-395) uncomfortably squeezed between Fig.s 16 & 17 and shift them after Fig.17.
- In line 405 at p.15 the Authors state that a "satisfactory" yield was obtained for the final bonding step used to finalize the integartion of all 5 building bolocks together in the final structure. However, this is a very fuzzy and non-specific and non-technical formulation. The Authors should instead clearly indicate how much is the yield (even if only approximately): 50%? 5%? 1%? 0.1? 0.001%?
Likewise, in lines 431-432 at p.17 it is stated that "high" bonding yields were obtained. What means "high"? What value(s) would qualify as "high"?
- At p.17, shift Fig.20 to be placed right under Fig.19, i.e. the lines of text in-between Fig.s 19 & 20 should be shifted down, to join the main bulk of the text under the Fig.s.
- The Literature Review probably could be improved to mention more of similar/related previous realizations reported earlier 9e.g. to refer to 1 or 2 examples for each type of component integrated in the system), but that may also increase the already quite significant length of the paper.
Once these corrections/additions are done, I believe that the Authors can forward the revised paper directly to the Editor for faster publication.
With best wishes,
The reviewer
Author Response

(The authors gave the same response as above.)

Reviewer 4 Report
Review of Micromachines-454239
The technological work described in this manuscript is very interesting.
So, from of the point of view of the technological results, it is quite a good work, though it is not clear what are the really innovative parts with respect to previous (and numerous) publications of the same authors on similar topics.
Their previous work is extensively cited in the text but it is easy to find also other publications of the same authors with quite similar figures. Actually, the authors should more clearly highlight what are the novelties included in the submitted manuscript compared to their previous papers.
I think the authors could take better care of the overall presentation of their work.
First of all, their reference list contains 14 publications and only 3 of them are from authors outside their own research group. I think these are some publications of other authors that are somehow related to this work and could be cited/commented in the introduction and added in the list.
https://doi.org/10.3390/mi9050219
https://doi.org/10.3390/proceedings2131067
https://doi.org/10.3390/mi8040126
https:// doi.org/10.1109/JSTQE.2014.2369499
https://doi.org/10.1117/1.JMM.13.1.011114
https://doi.org/10.1109/TRANSDUCERS.2017.7994036
A second remark is on the quality of their figures. The majority of them contains very small details that are very difficult to appreciate. Several text inserts in the figures are unreadable since realized with ultra-small characters. So, the authors are invited to improve their figures with regard to these issues.
Last, I would remove from the paper title “..for on-chip microscopy” since I do not find in the manuscript the demonstration of the application of their structure for generating microscopy images of any kind.
Minors:
Line 55 it should be micro beam-splitter, and not micro-beam splitter
Line 177 DRIE process not DRIE etch
Line 229 3.2. D MEMS, Why D?
Line 203, Figure 8a, The outcoming beam 1 does not come directly from the incoming beam but form the sample: I do not see this explanation in the text
Author Response

(The authors gave the same response as above.)
